# Alert: Learning Trigger Functions for Early Classification of Time Series using Deep-RL

## Abstract

Early Classification of Time Series (ECTS) is vital in fields like industrial monitoring and medical triage, where quick and accurate predictions are essential. The core challenge lies in the trigger function, which decides when to make a prediction, independently from the classifier itself. Most existing methods rely on handcrafted rules, but can data-driven approaches outperform them? This paper introduces ALERT, a Deep RL framework that learns trigger functions from any state representation. Systematic comparisons on 30 datasets show that the design of the state space significantly influences performance. Building on this, we propose ALERT$^+$, a simple yet effective variant that consistently outperforms traditional methods in balancing accuracy and delay, while significantly reducing empirical risk. ALERT and ALERT$^+$ are released to support reproducible research and practical applications.

## 1 Introduction

Many data-driven decision-making systems operate under *time pressure*, where delaying a prediction can be almost as costly as making an incorrect one. This tension is particularly acute in applications such as real-time monitoring of industrial assets, predictive maintenance, or medical triage, where acting too late may lead to equipment failure, safety incidents, or adverse clinical outcomes. In such settings, the central question is not only *what* to predict, but also *when* to issue a prediction.

Early Classification of Time Series (ECTS) (Gupta et al., 2020; Akasiadis et al., 2024; Renault et al., 2025) explicitly addresses this joint optimization of predictive accuracy and decision timeliness, and belongs to the broader class of early decision-making (Bondu et al., 2022) and anytime prediction problems. A widely adopted and practically attractive way to tackle ECTS is through *separable architectures*: a time-series classifier produces a sequence of probabilistic predictions over time, and a *trigger function* decides when the current prediction should be released. This separation is appealing because it decouples the design of powerful classifiers (Bagnall et al., 2017; Ruiz et al., 2021) from the design of decision policies that trade off misclassification and delay costs.

Most existing separable ECTS methods rely on carefully engineered trigger functions. These range from simple confidence thresholds and stopping rules (Mori et al., 2017; Schäfer and Leser, 2020), to more involved cost-based or non-myopic strategies that estimate future misclassification and delay costs (Dachraoui et al., 2015; Achenchabe et al., 2021; Tavenard and Malinowski, 2016). While often effective, these approaches are typically predefined and inflexible: they embed strong assumptions about how confidence, time, or expected costs should interact, and they are tailored to specific choices of features (e.g., maximum posterior, margin, or time index). As a result, they raise two key questions:

1. Can a *data-driven decision policy* systematically outperform handcrafted triggers when given access to the same information?

2. How should the *state representation* fed to such a policy be designed in order to leverage the information produced by a classifier most effectively?

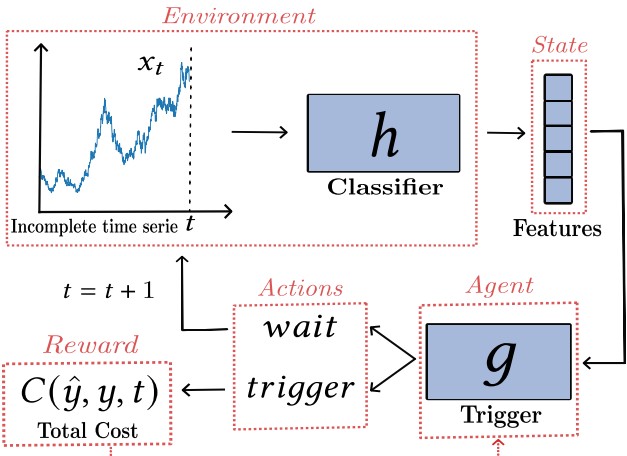

Figure 1: Separable ECTS pipeline and corresponding RL terminology (written in red).

Reinforcement Learning (RL) offers a principled framework for sequential decision-making under uncertainty, and appears particularly well-suited to ECTS (see Figure 1 for a terminology mapping). At each time step, a triggering agent must decide whether to *wait* for additional observations, incurring further delay costs but potentially improving future classification accuracy, or to *stop* and commit to the current prediction, locking in a misclassification cost if the prediction is wrong. This naturally matches the RL formalism, with the state space capturing information about the time series and classifier outputs, actions being {*wait*, *trigger*}, and rewards reflecting the combined misclassification and delay costs.

However, prior RL-based approaches to ECTS are typically end-to-end (Martinez et al., 2018; 2020; Wang et al., 2024; Hartvigsen et al., 2019) and tightly couple the triggering policy with a specific encoder or classifier (Hartvigsen et al., 2020; 2022; Huang et al., 2022a;b). This coupling makes it impossible to isolate the impact of the trigger policy itself, and leads to inherently biased comparisons.

In this paper, we bridge this gap by studying RL for *separable* and *classifier-agnostic* ECTS. We introduce **Alert** (**A** reinforcement **L**earning-based **E**arly classifie**R**'s **T**rigger function), a generic Deep RL framework or learning trigger functions that operate on arbitrary state representations derived from classifier outputs, and potentially on the incoming time series itself. This design enables controlled and fair comparisons, in which RL-based triggers and state-of-the-art handcrafted methods are given exactly the same input information. Our contributions are threefold:

**(C1) A generic RL framework for separable, classifier-agnostic ECTS.** We propose ALERT, a Deep RL framework that learns trigger functions on top of any pre-trained classifier, and enables controlled comparisons where RL-based and handcrafted separable methods operate on exactly the same input features and cost functions.

**(C2) A systematic study of state space design for RL-based triggers.** Across 30 datasets, we vary the state representation and observe substantial performance differences between alternative designs. This analysis clarifies which signals tend to be most informative for early decision policies and how they interact with different triggering mechanisms.

**(C3) Alert$^+$: a simple, robustly superior RL-based trigger.** By combining a small set of standard ECTS features, we obtain ALERT$^+$, which consistently attains the best average rank across our benchmarks and significantly reduces empirical risk compared to the strongest separable methods on a majority of datasets (see Section 5 for statistical tests). A Pareto analysis over the two conflicting objectives of prediction accuracy and earliness further shows that ALERT$^+$ dominates competitors on a large portion of the misclassification–delay trade-off spectrum.

Beyond raw performance improvements, ALERT serves as a unified and modular testbed for ECTS. By explicitly separating the classifier from the triggering policy, it allows us to attribute performance gains either to better predictors or to better decision rules, and to systematically study how state representations, reward designs, and training schemes impact results. The released ALERT implementation available at `https://anonymous.4open.science/r/ALERT-0066` is intended both as a reproducible research tool and as a practical asset for practitioners seeking to tune trigger functions for their own time series applications.

The remainder of the paper is organized as follows. Section 2 reviews background on RL in the context of ECTS and summarizes separable ECTS methods, with a focus on the information used to trigger decisions. Section 3 introduces the ALERT framework and discusses its components: state space, reward design, offline training, and model selection. Section 4 details the experimental protocol, including datasets, cost functions, and baselines. Section 5 presents the empirical results, analyzing the impact of state space design and comparing ALERT+ against state-of-the-art methods. Section 6 concludes with a discussion of implications and directions for future RL-based early decision-making systems.

## 2 Background

In ECTS, measurements of an input time series are observed over time. At any time $t < T$, before the last time step $T$, the incomplete time series $\mathbf{x}_t = \langle x_1, \ldots, x_t \rangle$ is available where $\{x_i\}_{(1 \leq i \leq t)}$ denotes the time indexed measurements (single or multi-valued). Each input time series belongs to an unknown class $y \in \mathcal{Y}$. The task is to make a prediction $\hat{y} \in \mathcal{Y}$ ideally at a time $t^\star \in [1, T]$ which optimizes a trade-off between: *(i)* the *misclassification cost* $C_m(\hat{y}|y) : \mathcal{Y} \times \mathcal{Y} \to \mathbb{R}$ of predicting $\hat{y}$ when the true class is $y$; *(ii)* the *delay cost* $C_d(t) : \mathbb{R}^+ \to \mathbb{R}$ of releasing the prediction at $t \in [1, T]$.

This section presents the background related to both RL and ECTS. Specifically, in Section 2.1, we begin by discussing the use of RL in a separable ECTS context before examining the existing literature on RL for ECTS. Then, in Section 2.2, separable ECTS methods are discussed in light of the input information they use to trigger decisions.

### 2.1 Adapting RL to ECTS

Reinforcement learning (Sutton and Barto, 2018) aims to learn a function, called a *policy $\pi$*, from states to actions: $\pi : \mathcal{S} \to \mathcal{A}$. *Rewards* can be associated with transitions from states $s_t \in \mathcal{S}$ to states $s_{t+1} \in \mathcal{S}$ under an *action $a \in \mathcal{A}$*: $reward(s_t, a, s_{t+1}) \in \mathbb{R}$. Given a state $s_t$ and an action $a_t$, the sequence of rewards received after time step $t$ gives rise to a *gain* which classically is a discounted sum of the rewards: $G_t = \sum_{k=0}^{\infty} \gamma^k r_{t+k+1}$ with $\gamma \in [0, 1]$ the discount factor and $r_t = reward(s_t, a_t, s_{t+1})$. In all generality, the result of an action $a$ in state $s_t$ may be non-deterministic. An optimal policy $\pi^\star$ maximizes the expected gain from any state $s_t \in \mathcal{S}$.

In the context of separable ECTS, the agent must learn which *action* to take (i.e. decision "*wait*" or "*trigger*" prediction $\hat{y}$) given its current *state*, i.e. any information provided about the incoming time series $\mathbf{x}_t$ and/or the classifier. An *episode* is defined as the sequence of states $s_t$ and actions $a_t$ starting from $t = 1$ until a prediction is triggered, and necessarily before the deadline $T$.

To our knowledge, there are no RL-based ECTS approaches in the literature that are both *separable* and *classifier-agnostic*, which is the scope of this paper. However, this is necessary for a fair comparison with the wide range of state-of-the-art approaches not based on RL. The aim of this paper is to bridge this gap in the literature by presenting a well-founded approach to building such systems. The remainder of this section presents the RL-based approaches in the literature, specifying whether they are separable or not, and which reward function is used.

**Review of RL-based ECTS approaches:** Martinez et al. (2018; 2020) exploit value-based RL in an end-to-end fashion. In any non-separable approaches: *(i)* the RL agent either decides to wait or makes a class prediction and therefore the action space is $\mathcal{A} = \{wait, y_0, \ldots, y_K\}$ if there are $K$ classes; *(ii)* the state space is defined as the time series themselves. In this approach, the reward function used penalizes the action of waiting and gives equal importance to correct and incorrect predictions.

EarlyStop-RL (Wang et al., 2024) is a cost-sensitive end-to-end approach, alluding to the possibility of considering any cost function. Indeed, the reward directly corresponds to any misclassification cost and any incremental delay cost, which combine additively.

EARLIEST and its variants use policy gradient algorithms (Hartvigsen et al., 2019; 2020; 2022) and consist of distinct modules (i.e. an encoder, a triggering agent, and a classifier), each of which operating with its own set of parameters. Thus, these approaches are *separable* even if these modules are jointly trained, but they are *not classifier-agnostic* by design. The reward function used only takes into account the misclassification error, the delay cost being a part of the loss function.

The SNP approach (Huang et al., 2022a;b) is a *separable* architecture that consists of ($i$) a specific classification module including an encoder; ($ii$) a RL agent that triggers predictions based on the embedding given by the classification module. This approach is *not classifier-agnostic* by design, and uses a specific reward to favor early and correct decisions with a logarithmic weighting on the trigger time.

## 2.2 SOTA: insights from non-RL separable ECTS

PROBA THRESHOLD (Renault et al., 2025) is a basic method monitoring the highest conditional probability estimated by the classifier $\max_{y \in \mathcal{Y}} p(y|\mathbf{x}_t)$, which is a simple measure of classifier confidence over time. As soon as it exceeds a value, which is a hyperparameter of the method, a prediction is made.

Rather than only considering the largest posterior, SR (Mori et al., 2017) considers three input variables to trigger decisions $(p_1, p_2, \frac{t}{T})$. The output of the system is defined as:

$$
g(h(\mathbf{x}_t)) = \begin{cases} \hat{y} = \underset{y \in \mathcal{Y}}{\arg\max}\, p(y|\mathbf{x}_t) & \text{if } \gamma_1\, p_1 + \gamma_2\, p_2 + \gamma_3 \frac{t}{T} > 0 \\ \emptyset & \text{otherwise} \end{cases}
$$

where $p_1$ is the largest posterior probability $p(y|\mathbf{x}_t)$ estimated by the classifier $h$, $p_2$ is the margin i.e. the difference between the two largest posterior probabilities, and $\frac{t}{T}$ represents the proportion of the incoming time series at time $t$. The parameters $\gamma_1$, $\gamma_2$ and $\gamma_3$ are learned from the training set.

The TEASER method (Schäfer and Leser, 2020) uses a classifier $h$ that computes the class probabilities for the incoming $\mathbf{x}_t$ and transmits them to the trigger function. In turn, $g$ is a classifier in its own right that classifies the prediction $\hat{y}_t$ as reliable or not. Finally, the trigger function takes the last reliable predictions and their associated times and decides to trigger the prediction $\hat{y}$ only if the same prediction was also given for a number of successive time steps (a hyperparameter to be tuned).

ECONOMY and its variant (Dachraoui et al., 2015; Tavenard and Malinowski, 2016; Achenchabe et al., 2021; Zafar et al., 2021) are non-myopic approaches, where the trigger function computes the expected costs, a combination of the misclassification cost and the delay cost, for the current time step $t$ and all future ones. A decision is triggered as soon as the expected cost for the current time step $t$ is the lowest among all future times expected costs.

CALIMERA (Bilski and Jastrzebska, 2023) is another non-myopic approach that exploits a collection of regressor models, learned in a backward induction fashion as a trigger function. The minimum expected cost occurring in the future time period $[t, T]$ is denoted as $minFutureCost_t$. For a particular time step $t$, the corresponding regressor aims to predict the difference $\Delta = minFuturCost_{t+1} - Cost_t$. Then, CALIMERA triggers a decision when $\Delta > 0$, i.e. the optimum trigger time is about to be exceeded. The collection of regressors is trained using the following variables: ($i$) all posterior probabilities; ($ii$) the maximum posterior $p_1$; ($iii$) the margin $p_2$.

Other separable ECTS approaches include (Lv et al., 2019; 2023; Schäfer and Leser, 2020; Ebihara et al., 2023). For a more comprehensive description of the ECTS literature, including non-separable approaches, the reader can refer to the survey of Renault et al. (2025).

# 3 Proposed methodology

In this section, we present ALERT: a RL-based separable ECTS framework that allows experimenting with various state spaces, reward functions, and classifiers. Its components are described, and possible choices for the state space and the reward functions are discussed.

## 3.1 Alert's pipeline overview

ALERT is a Deep RL trigger function, that can be used in any kind of separable ECTS architecture. Below, we first outline the main components of ALERT:

- **The state space and reward description**: The first step in designing a RL-based trigger function is to define the state space and what the reward function should be. That is, the inputs of the trigger model and the function to be maximized in order to achieve an effective policy.

- **The experience buffer generation**: Once the setup has been defined, ALERT exhaustively generates all possible interactions by simulating "wait" or "trigger" actions for every possible time step, for every training sample.

- **The offline DDQN training**: At this stage, a DDQN model is trained in an offline fashion, i.e. solely based on the buffer generated in the previous step.

- **The *post hoc* model selection**: Finally, a key component for ALERT to work effectively across a various range of different environments is the model selection phase. In this final step, one model is selected from those learned during training to avoid overfitting and improve method stability.

## 3.2 State space

In RL, the state, together with the reward signal, provides all the information that the agent uses to decide which next action to take. In the context of ECTS, the state space may include any information from the classifiers as well as the time series themselves. It is desirable that the components of the state space be relevant to the classification task to be performed, while remaining limited in number. This compromise is of great importance in the design of effective and efficient approaches (Curran et al., 2015).

Throughout this article, several sets of candidate components of the state space vector will be considered, mainly inspired by the most efficient state-of-the-art algorithms (Renault et al., 2025): (*i*) the *predicted class label* (Schäfer and Leser, 2020): $\arg\max_{k\in\mathcal{Y}} p(y=k|x_t)$; (*ii*) the *maximum posterior* (Mori et al., 2017; Bilski and Jastrzebska, 2023): $\max_{k\in\mathcal{Y}} p(y=k|x_t)$; (*iii*) the *margin* (Mori et al., 2017; Schäfer and Leser, 2020; Bilski and Jastrzebska, 2023) which is the difference between the two largest posterior probabilities; (*iv*) the estimation of the *level of confidence* (Achenchabe et al., 2021; Zafar et al., 2021) in the prediction(s), which can be represented by the bin index within an equal-frequency discretization of maximum posteriors; (*v*) the *current time t* (Mori et al., 2017), which, in the case of finite time series of length $T$, provides the proportion of the time series observed so far; (*vi*) the *time series* itself, sub-sampled or not.

Several combinations of the mentioned candidates have been tested as state spaces (see Section 5.1). All of the components of the state space have been normalized when needed, to keep them all in the $[0,1]$ range, as suggested by Tarasov et al. (2024); Fujimoto and Gu (2021). Particularly, the predicted class label has been one-hot encoded and the indexes of confidence levels have been MinMax scaled.

## 3.3 Reward function

There can be multiple choices, organized according to two main options : **(1)** provide the reward at the end of each episode only, or **(2)** provide rewards at every time step. The rest of this section describes these two types of rewards.

**(1)** Reward is provided at the *end of episodes.* The most straightforward approach is to provide a reward directly related to the paid cost at the time the decision is made:

$$reward^{trigger} = -\mathrm{C}(\hat{y}_t, y, \hat{t}) = -\mathrm{C}_m(\hat{y}_t|y) - \mathrm{C}_d(\hat{t}) \tag{1}$$

**(2)** Rewards are provided at *every time step*[1]. At each time step, the agent can receive information that waiting entails a delay cost:

$$reward^{wait} = -\Delta\mathrm{C}_d(t) = \mathrm{C}_d(t) - \mathrm{C}_d(t-1), \quad \text{with } \mathrm{C}_d(0) = 0 \tag{2}$$

In both latter cases, the final opposite classification error cost $-\mathrm{C}_m(\hat{y}_{\hat{t}}|y)$ is provided at the time the prediction $\hat{y}_{\hat{t}}$ is made. Alternative rewards functions, inspired by imitation learning (Hussein et al., 2017; Kumar et al., 2022), that depend on optimal trigger time $t^\star$ or associated cost $\mathrm{C}^\star$, are investigated in Appendix D.2.

### 3.4 Interactions buffer

Given a complete training time series $\mathbf{x}_T$, it is possible to compute for each time step $t \in \{1, T\}$ the cost of making a decision: $\mathrm{C}_m(\hat{y}_t|y) + \mathrm{C}_d(t)$. Given $N$ training time series, this can potentially yield $N \times T$ training examples for learning the trigger function. However, as usually done in the literature (Mori et al., 2017; Schäfer and Leser, 2020; Bilski and Jastrzebska, 2023), the time series are divided into a smaller, limited number of time step, rather than considering all time steps $t$. This significantly reduces the number of possible transitions and provides a way to standardize the number of time steps considered when datasets contain time series of different lengths. Typically, the number of time steps considered is set to 20, each one representing 5% of the time series. For each of these 20 time steps, both the "*wait*" and the "*trigger*" actions are simulated, and the associated rewards are computed, using pre-trained classifier's predictions.

### 3.5 Offline DDQN

Finally, a Double Deep-Q-Network (DDQN) model (Mnih et al., 2015; Van Hasselt et al., 2016) is learned in an offline manner to act as our triggering agent. It is based on the classical *Q-learning* (Dayan and Watkins, 1992) algorithm.

**Q-Learning:** The algorithm consists of learning a policy by using a state-action value function, which, for each pair, maps $(s, a)$ the expected gain starting from $s$, taking action $a$ and following $\pi$ afterwards:

$$q_\pi(s_t, a_t) \doteq \mathbb{E}_\pi[G_t \mid s_t, a_t] \tag{3}$$

where $\mathbb{E}_\pi[\cdot]$ denotes the expected value of a random variable given that the agent follows the policy $\pi$.

Optimal policies share the same optimal action-value functions:

$$\forall (s_t, a_t) \in \mathcal{S} \times \mathcal{A}: \quad q^\star(s_t, a_t) = \max_\pi q_\pi(s_t, a_t) \tag{4}$$

Thus, provided that an optimal state-action value function has been found, an optimal policy can be derived: $\pi^\star(s) = \arg\max_a q^\star(s, a)$. Q-learning is a popular way of directly approximating $q^\star$ with updates:

$$Q(s_t, a_t) \leftarrow Q(s_t, a_t) + \delta\big[r_t + \gamma \max_a Q(s_{t+1}, a) - Q(s_t, a_t)\big] \tag{5}$$

where $Q$ is the estimated $q$ function, $r_t$, the immediate reward and $\delta$ the learning rate.

When working with high-dimensional and/or continuous state-action spaces, one typically learns a parametrized value function $Q(s, a, \theta)$ (Mnih et al., 2015) where $\theta$ should minimize the following:

$$\mathcal{L}(\theta|s_t, a_t, r_t, s_{t+1}) = \Big(\overbrace{\big(r_t + \gamma \max_{a_{t+1}} Q_\theta(s_{t+1}, a_{t+1})\big)}^{Y^{DQN}} - Q_\theta(s_t, a_t)\Big)^2 \tag{6}$$

---

[1]Note that this formulation assumes that misclassification and delay costs can be decomposed additively.

In practice, for stability reasons, the Q-learning target $Y^{DQN}$ in Deep Q-Network (DQN) is computed via a target network, with parameters $\theta^-$, equal to those of the main network, except that its parameters are governed by an exponential moving average of the main network's parameters, with a smoothness parameter $\tau$ (Lillicrap, 2015).

One well-known issue in DQN is the overestimation bias, which often occurs as the standard DQN update rule is based on a maximum value of a noisy and uncertain estimation of the Q-values, potentially accumulating error as training goes on. *Double DQN* (Van Hasselt et al., 2016) addresses it by de-correlating action selection from action evaluation. In practice, one can simply use the main online network to select the action, then to be evaluated using the target network. The DDQN target thus becomes:

$$Y^{DDQN} = r_t + \gamma Q_{\theta^-}(s_{t+1}, \arg\max_a Q_\theta(s_{t+1}, a)) \tag{7}$$

**Offline DDQN :** ALERT adopts an *offline* training scheme (Levine et al., 2020; Kostrikov et al., 2021). This reduces the computational burden, and allows training the triggering agents over many different datasets. In order to further compare ALERT with state-of-the-art algorithms, a full batch training procedure has been chosen for all competitors, thus including ALERT. Note that extending ALERT to offline-to-online or fully online settings remains possible. In addition, some disadvantages of Offline RL, particularly the sensitivity to distribution shifts, appear somewhat mitigated when approaching ECTS as done in this paper. Indeed, as mentioned earlier, the static buffer is exhaustive and balanced in terms of action representation. Thus, out-of-distribution actions are not a matter here, reducing extrapolation error risk.

Transitions are sampled uniformly, as done by Cetin et al. (2024), from the buffer $\beta$ and transmitted to the Q-estimator neural net in order to minimize the loss in Equation 7. We used Layer Normalization (Lei Ba et al., 2016) in the Q-network since it further mitigates overestimation bias by bounding the outputted Q-values (Ball et al., 2023; Tarasov et al., 2024). Once trained, the Q-Network is used as our triggering module: based on an observed state $s_t$, action $a_t$ is selected such that its associated estimated Q-value is maximal: $a_t = \arg\max_{a \in \mathcal{A}} Q(s_t, a)$.

### 3.6 Model selection

Regularization is ensured by a model selection strategy that limits the number of epochs and thus effectively combats overfitting. First, several policies are trained over different train/validation splits, as this greatly improves offline off-policy evaluation (Nie et al., 2022). Then, for each split, the policy under training is evaluated over the validation set at a given epoch frequency in an *online* fashion, i.e. with iterative measurements over the time series, like at testing time. The *AvgCost* is used as a metric at each validation stage, and the corresponding model is saved. Finally, after training, the epoch index for which the validation metric is lowest on average across all splits is selected. Among the models trained for this number of epochs, the best performing one over all splits is selected to be the final model.

## 4 Evaluation protocol

In the experiments conducted, the following state-of-the-art algorithms were tested: PROBA THRESHOLD, STOPPING RULE, ECONOMY, CALIMERA and EARLIEST. More details about the competitors and the experimental protocol, including datasets and implementation specifications, are reported in Appendix A.

To meet the requirements of potential applications such as anomaly detection or decision-making in hospital emergency departments, we have chosen to use unbalanced misclassification costs and exponential delay costs, which are more consistent with real-world conditions, as done by Renault et al. (2025). For our experiments, we used the definition of the costs described:

$$C_d(t) = \exp(\frac{t}{T} \times \log 100) \tag{8}$$

$$C_m(\hat{y}|y) = \begin{cases} 100 \times \mathbb{1}(\hat{y} \neq y) & \text{if } y \in \text{minority class} \\ \mathbb{1}(\hat{y} \neq y) & \text{otherwise} \end{cases} \tag{9}$$

Alternative cost functions are evaluated in Appendix B.5. Evaluation is conducted using the *AvgCost* metric. Furthermore, in order to assess how the methods adapt to various balances $\alpha$ between the misclassification and the delay costs, the evaluation is done using a weighted *AvgCost*, for a set of $N$ labeled time series, for values of $\alpha$ varying from 0 to 1, with a 0.1 step:

$$AvgCost_\alpha = \frac{1}{N} \sum_{i=0}^{N} \alpha \times C_m(\hat{y}_i|y_i) + (1 - \alpha) \times C_d(\hat{t}_i) \tag{10}$$

## 5 Experimental results

The ALERT framework presented in Section 3 is used in our experiments to learn various trigger functions that exploit different state spaces and reward functions. The experiments reported in this section aim to shed light on the two questions:

1. First, *given the same input information, how do we compare the performance of handcrafted trigger functions and their RL counterparts*? Section 5.1.1 examines this question and the impact of state space design. Using this comparison, new state spaces can be tested and their performance measured. Section 5.1.2 considers different trigger functions learned from several variations of the state space and concludes by highlighting a new RL-based system, called ALERT⁺.

2. *How does* ALERT⁺ *fare against state-of-the-art approaches*? (see Section 5.2)

In this section, the used reward function is Equation 2, i.e. distilling temporal cost at each time step.

Table 1: State spaces of ALERT's variants.

| | Max. posterior | All posteriors | Margin | Pred. class | Lvl of conf. (num. bins = 10) | Time | Time Series (num. of points) | Total Dim $|S|$ |
|---|---|---|---|---|---|---|---|---|
| ALERT PT | ✓ | | | | | | | 1 |
| ALERT ECO | | | | ✓ | | | | 1 |
| ALERT SR | ✓ | | ✓ | | ✓ | | | 3 |
| ALERT POST. | | ✓ | | | | | | $|\mathcal{Y}|$ |
| ALERT CAL | ✓ | ✓ | ✓ | | | | | $|\mathcal{Y}| + 2$ |
| ALERT⁺ | ✓ | | ✓ | ✓ | ✓ | ✓ | | $|\mathcal{Y}| + 4$ |
| ↪ SUB SERIES[5] | ✓ | | ✓ | ✓ | ✓ | ✓ | ✓(20) | $|\mathcal{Y}| + 24$ |
| ↪ FULL SERIES | ✓ | | ✓ | ✓ | ✓ | ✓ | ✓(T) | $|\mathcal{Y}| + 4 + T$ |

### 5.1 State space design

### 5.1.1 Same state space: SOTA vs. Alert

Here, we compare the performance of the handcrafted trigger functions of three representative algorithms: ECONOMY, SR and CALIMERA for ECTS, and the corresponding RL-based trigger functions: ALERT ECO, ALERT SR and ALERT CAL, which use the same input information (see Table 1).

Figure 2 provides an overall qualitative view of the performance for each handcrafted trigger function vs. RL-based one for values of $\alpha \in [0, 1]$. The overall trend is that **as the state space gets enriched, RL-based algorithms tend to outperform more significantly their handcrafted counterparts** thus suggesting that RL is able to better extract trigger functions. In effect, ECONOMY's trigger function extracts

---

[5] Time series are averaged over windows, representing 5% of the series, so that the resulting number of points is equal to 20.

from $\mathcal{S} = \{level\ of\ confidence\}$ what turns out to be useful for decision. For instance, in Figure 2a, when $\alpha = 0.8$, ECONOMY wins over almost 85% of the datasets, and is 10% closer to the $AvgCost^\star$. This is clearly less the case when operating over larger state space $\mathcal{S} = \{all\ posteriors, max\ posterior, margin\}$.

One related question naturally arises: is this enough to enlarge the state space to get better RL-based trigger functions? A set of experiments using ALERT has been designed to answer these questions.

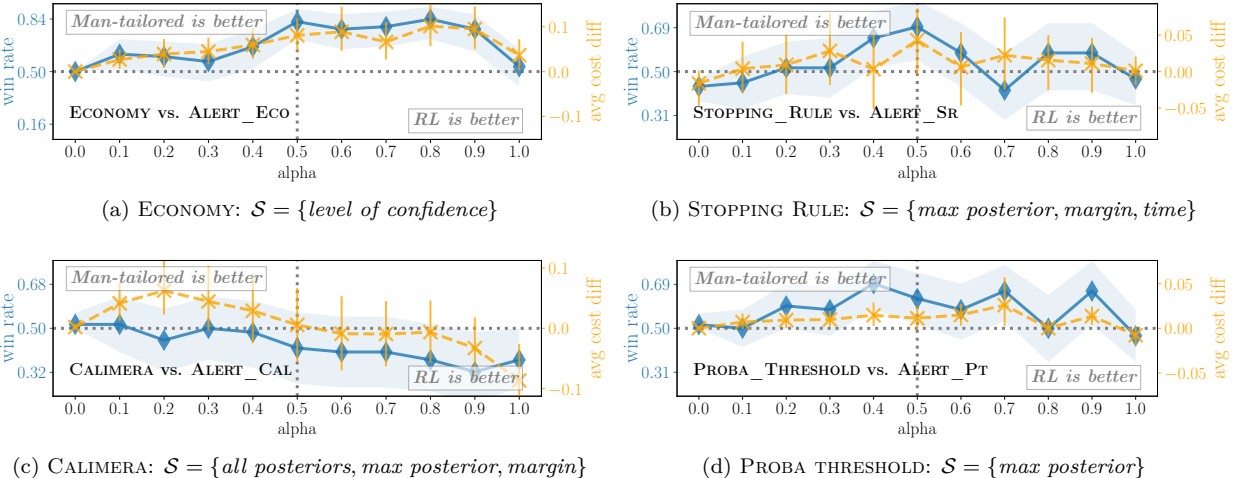

(a) ECONOMY: $\mathcal{S} = \{level\ of\ confidence\}$

(b) STOPPING RULE: $\mathcal{S} = \{max\ posterior, margin, time\}$

(c) CALIMERA: $\mathcal{S} = \{all\ posteriors, max\ posterior, margin\}$

(d) PROBA threshold: $\mathcal{S} = \{max\ posterior\}$

Figure 2: Pairwise comparison of SOTA methods versus their RL counterpart using same information as input. The blue curve, ranging from 0 to 1, represents the win rate of the original method. The orange curve, ranging from -1 to 1, represents the difference of $AvgCost$ between SOTA and RL counterparts, normalized by $AvgCost^\star$. In both cases, points above the horizontal line indicate that the original method is better than its RL-based counterpart. Shaded areas/vertical bars correspond to 90% bootstrap confidence intervals.

### 5.1.2 Exploring the design of state spaces for RL-based trigger functions:

One advantage of ALERT is that adding features to the state space has little impact on computation time and implementation. It is thus easy to try out a number of variations of the state space used. Specifically, we compare the performance obtained with ALERT using the state spaces displayed in Table 1, introducing ALERT$^+$ which relies on the union of state spaces of the best SOTA approaches, namely: CALIMERA, ECONOMY, SR and TEASER.

Figure 3 reports the ranking of the resulting ECTS performance on the datasets while testing the versions of ALERT in the upper part of Table 1. A clear trend emerges, with a marked advantage when ALERT uses larger state spaces. In addition, it is interesting to see that ALERT POSTERIORS is outranked both by ALERT$^+$ and by ALERT CAL, while it is possible to derive the inputs of these two algorithms using the inputs of the former, emphasizing the impact of processing input information. Additional results about each tested features' importance can be found in Appendix D.1.

Instead of considering only features used in handcrafted trigger functions from the literature, *could it be that by further increasing the state space, better performance still can be reached?* To answer that question, we tried including even more information, related to the time series, in the state space and compared them with ALERT$^+$ (lower part of Table 1). This experiment shows that even though ALERT$^+$ has the most limited state space in size in this comparison, it tops the versions that somehow include time series information (see Appendix B.2 for more details).

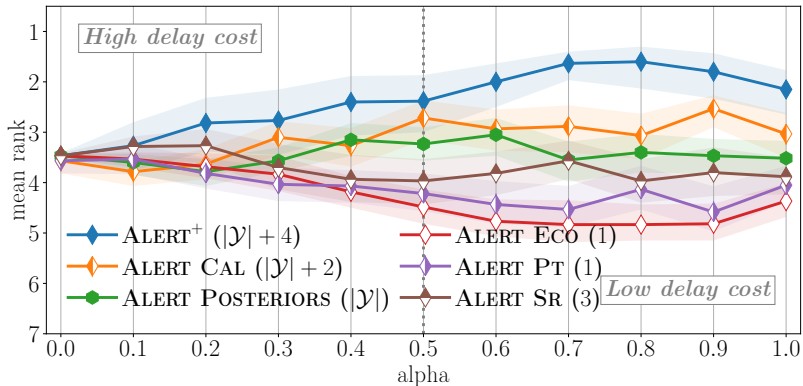

Figure 3: Evolution of the mean ranks, for every $\alpha$. ALERT$^+$ is combining SOTA state spaces.

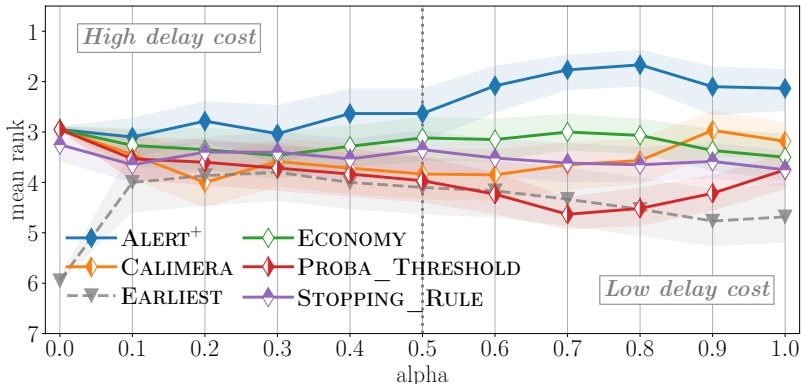

(a) Comparison of the mean ranks, for every $\alpha$, based on the *AvgCost* metric. Shaded areas correspond to 90% bootstrap confidence intervals.

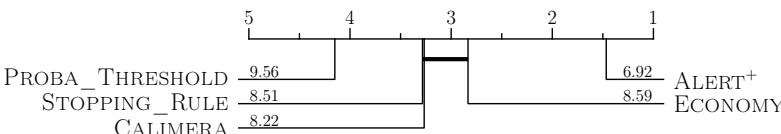

(b) Wilcoxon signed-rank test labeled with mean *AvgCost*, $\alpha = 0.8$.

Figure 4: The ranking plot (a) shows that, across all $\alpha$, ALERT$^+$ dominates all competitors. This result is significant as supported by statistical tests as shown in plot (b) for $\alpha = 0.8$.

## 5.2 Alert$^+$ vs. State-of-the-art

The previous section concluded that ALERT$^+$ seems to strike a good balance between too small a state space and too a large one with no gain in useful information. But how does it fare against the state-of-the-art competitors described in Section A.2?

**Comparison in terms of ranks and *AvgCost*:** The results of the experiments carried out are summarized in Figure 4. First of all, the new system ALERT$^+$ outranks all competitors for all values of $\alpha$ on average on the datasets. Second, looking at *AvgCost*, what is paid actually when using the algorithms, ALERT$^+$ again significantly outperforms its competitors, for example, when $\alpha = 0.8$ as shown with the Wilcoxon signed-rank-test, that is for low delay costs. For high delay costs, such as exponential ones, the superiority is less pronounced, as the optimal policy is to decide early on, which all algorithms tend to do.

**Comparison using Pareto front:**  Considering the minimization of misclassification and delay costs as two conflicting objectives, one can draw the Pareto front for each method: the set of points for which no other point dominates with respect to both objectives. Figure 5 shows the result for $\alpha \in \{0, 0.1, 0.2, \ldots, 1.0\}$. What stands out first is the clear domination of ALERT$^+$. A closer examination reveals that ALERT$^+$ generally makes its decision later than its competitors, risking higher delay costs, but this extra cost is generally more than offset by lower misclassification costs.

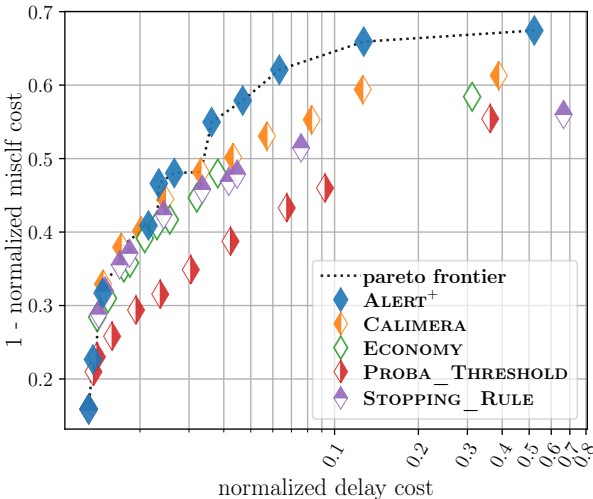

Figure 5: Pareto front, displaying for each $\alpha$, the normalized version of the *AvgCost*, decomposed over delay and one minus misclassification cost on $x$-axis and $y$-axis respectively. Best approaches are located on the top left corner. High $\alpha$ values are located on the right, low ones on the left. Due to the exponential shape of the delay cost, the $x$-axis is on log scale.

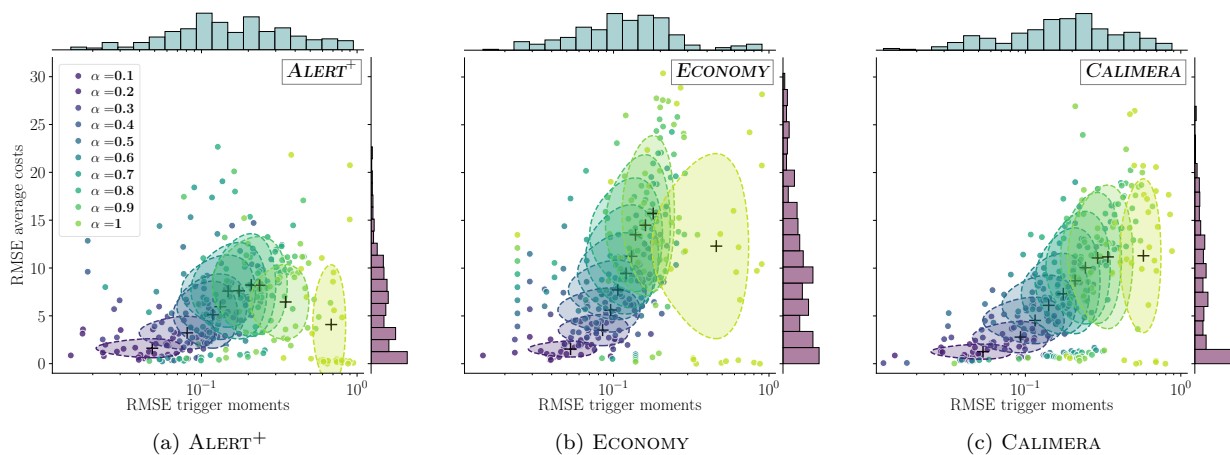

(a) ALERT$^+$                    (b) ECONOMY                    (c) CALIMERA

Figure 6: The $x$-axis reports how far is the normalized triggering time from the best *a posteriori* one, on log scale: left is better. The $y$-axis reports the difference between the *AvgCost* incurred by the algorithm compared to the best *a posteriori* one, *AvgCost*$^\star$: lower is better. The black crosses report the average performance for each value of $\alpha$ (greater values correspond to lower relative importance of the delay cost). Ellipses display $2\times$ the standard deviation over both axis, computed for each $\alpha$ value.

**Detailed comparison of Alert$^+$, Economy and Calimera:**  Figure 6 conveys a wealth of information (the rest of the competing approaches can be found in Appendix B.4). The $x$-axis reports how far the

normalized trigger time is from the best *a posteriori* one, on log scale, the smaller the better. The $y$-axis reports the difference between the *AvgCost* incurred by the algorithm compared to the best *a posteriori* one, *AvgCost$^\star$*, again lower is better. The colored dots correspond to experimental results for the datasets and for values of $\alpha \in \{0, \ldots, 1\}$. The marginals for the $x$-axis and the $y$-axis are reported resp. on the top and on the right of the graph.

It is readily apparent that ALERT$^+$ tends to yield a lower average cost, especially for intermediate values of $\alpha$, confirming the findings reported above. ALERT$^+$ is also more robust with respect to the variety of datasets, displaying smaller ellipses, thus less variance, than its competitors. Finally, ALERT$^+$ spreads its decision times more than its competitors. For low values of $\alpha$ (high delay costs), the ellipses of ALERT$^+$ are more extended along the $x$-axis: with some values of RMSE trigger moments half of the ones by ECONOMY and CALIMERA. For high values of $\alpha$ (low delay costs), ALERT$^+$ tends to spread its decision time (high RMSE trigger moments) even more than ECONOMY and significantly more than CALIMERA. It can indeed afford it in this regime. Overall, ALERT$^+$ seems to better handle the decision time in favor of improved performance in terms of *AvgCost*. Toward the better understanding of the ALERT$^+$ good performances, qualitative results are examined in Appendix E.

## 6 Conclusion

This work investigated whether Reinforcement Learning (RL) can provide tangible benefits in Early Classification of Time Series (ECTS). To address this question in a principled way, we focused on separable architectures and explicitly decoupled the time-series classifier from the triggering policy. We introduced ALERT, a generic Deep RL framework for separable, classifier-agnostic ECTS that learns trigger functions on top of any pre-trained classifier, from user-defined state representations. This design enables, to the best of our knowledge, the *first* controlled comparison of RL-based and handcrafted separable triggers under identical information.

Using 30 public datasets, we conducted a systematic study of state space design for RL-based triggers. Our experiments show that, given the same input information, RL agents can match or improve upon strong handcrafted trigger functions from the ECTS literature. By focusing on the impact of the information used by the decision-making function, this paper shows that an informed choice of descriptors can lead to significant performance gains. This choice itself can be considered a hyperparameter to be optimized according to the type of application and the data available, a perspective that is new to the ECTS field and open to easy investigation using the ALERT framework. Building on these insights, we derived ALERT$^+$, a simple instantiation that combines a small set of widely used ECTS features. ALERT$^+$ consistently attains the best average rank across a wide range of misclassification–delay trade-offs and, compared to the strongest separable methods, yields statistically significant reductions in empirical risk, while also achieving favorable Pareto fronts over misclassification and delay costs.

Our published open source implementation is intended both as a reproducible research tool and as a practical asset for time-critical applications, such as industrial monitoring, predictive maintenance, and medical triage.

Finally, one of the challenges facing ECTS is early classification in non stationary environments, either when covariate shift occurs (e.g. the prevalence of anomalies changes over time) and/or when there is concept shift (e.g. the same times series may point to a different type of fraud). Furthermore, in the context of ECTS, the prediction itself may alter the future of the time series. (e.g. when braking is triggered on an autonomous vehicle when a collision is predicted). In all these cases, the adaptive nature of RL to non-stationary environments seems to be an interesting avenue to explore.

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

## Appendix

## A  Experimental protocol

### A.1  Datasets

Extensive experiments have been carried out on 30, both synthetic and real-life datasets: 19 from the UCR archive (Dau et al., 2019) (both binary and multi-class) and $11^2$ from the Monash time series extrinsic regression archive (Tan et al., 2021), transformed into binary classification tasks. These are the same used as Renault et al. (2025) and can be downloaded from these links[3]. We selected datasets that are not z-normalized, to avoid possible information leakage (Wu et al., 2021).

### A.2  Competing Approaches

Four competing separable approaches have been selected from the top performers benchmarked by Renault et al. (2025). In addition, the end-to-end approach EARLIEST, although not directly comparable since it does not use the same classifier, is considered in our experiments because it is the main RL-based competitor from the literature.

- CALIMERA (Bilski and Jastrzebska, 2023) triggers a decision when the value predicted by the regressor models becomes positive.

- ECONOMY-$\gamma$-MAX (Achenchabe et al., 2021; Zafar et al., 2021) triggers a decision if the predicted cost expectation is the lowest at time $t$ when compared with the expected cost for all future time steps.

- STOPPING RULE (Mori et al., 2017) uses a linear combination of two confidence levels and a delay measure.

- PROBA THRESHOLD triggers a prediction if the maximum posterior exceeds some threshold, found by grid search.

- EARLIEST (Hartvigsen et al., 2019) is the only end-to-end Deep RL method that has been adapted, in our experiments, to different cost settings by cross-validating the $\lambda$ hyperparameter, measuring the importance of the earliness.

### A.3  Implementation specifications

For all datasets, we use 70% training, 30% testing. Within the training set, 50% is used for classifiers' and 50% for training the trigger model. The classifier module is a time-indexed set of MiniROCKET (Dempster et al., 2021) estimators, learned over every 5% of the time series. A calibration step is added as done by Bilski and Jastrzebska (2023); Renault et al. (2025). For ALERT models, 30% of the trigger training data is used for validation and model selection. We use a single-layer Neural Network with a hidden dimension of 32 to be our Q-estimator. The model is optimized using Adam (Kingma and Ba, 2015) with a learning rate of $1e^{-4}$. The target network uses soft updates based on parameter $\tau$ (Lillicrap, 2015) equal to $3e^{-3}$. The code to run the experiments is available on `https://anonymous.4open.science/r/ALERT-0066`. It is based on `PyTorch` (Paszke et al., 2019) for automatic differentiation and on `ml_edm` (Renault et al., 2024) for general ECTS evaluation functions and interface.

Table 2: Wilcoxon tests p-values, comparing pairwise man-tailored and RL-based algorithms. **Bold**, resp. *Italic* values indicate a value below the significance level equal to 0.05, in favor of man-tailored method, resp. RL-based method.

| $\alpha \rightarrow$
method $\downarrow$ | 0 | 0.1 | 0.2 | 0.3 | 0.4 | 0.5 | 0.6 | 0.7 | 0.8 | 0.9 | 1 |
|---|---|---|---|---|---|---|---|---|---|---|---|
| ECONOMY | NaN | 0.062 | 0.091 | 0.082 | **0.003** | **<1e-3** | **<1e-3** | **<1e-3** | **<1e-3** | **0.001** | 0.421 |
| STOPPING RULE | *0.046* | 0.672 | 0.701 | 0.635 | 0.15 | **0.01** | 0.134 | 0.931 | 0.111 | 0.141 | 0.673 |
| CALIMERA | 0.317 | 0.464 | 0.522 | 0.644 | 0.799 | 0.961 | 0.604 | 0.474 | 0.161 | 0.069 | 0.086 |
| PROBA THRESHOLD | 0.317 | 0.803 | 0.066 | 0.097 | **0.005** | 0.074 | 0.112 | **0.036** | 0.953 | **0.018** | 0.386 |

# B  Supplementary Results

## B.1  Pairwise comparison: statistical tests

## B.2  State space: adding time series

As a baseline, we enlarge ALERT$^+$ state space with twenty randomly drawn points (ALERT$^+$ RANDOM).

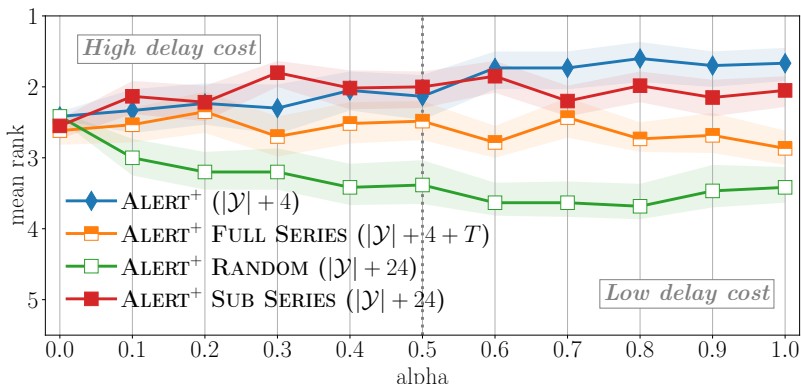

Figure 7: Evolution of the mean ranks, for every $\alpha$, based on the *AvgCost*. Adding time series on ALERT$^{+'}$ state space.

Figure 7 shows that even though ALERT$^+$ has the most limited state space in size in this comparison, it tops both ALERT$^+$ FULL SERIES and ALERT$^+$ RANDOM for all values of $\alpha$, and is better than ALERT$^+$ SUB SERIES for low delay cost ($\alpha \geq 0.6$) and equal or second for $\alpha \leq 0.5$. Furthermore, the size of the state space itself is not a predictor of performance, since ALERT$^+$ RANDOM is the last for all values of $\alpha$. The choice of input features plays a role in performance, as evidenced by the fact, with the same state space size, ALERT$^+$ SERIES clearly outperforms ALERT$^+$ RANDOM.

---

[2]And not 15 as done by Renault et al. (2025), as in the anomaly detection setting, some of the problems become too hard for the considered classifiers to operate, i.e. there is no performance gain when increasing the number of observations in the time series.

[3]http://timeseriesclassification.com/dataset.php, https://urlz.fr/qRqu

## B.3 Alert$^+$ vs. SOTA: statistical tests

Table 3: Wilcoxon tests p-values, comparing Alert$^+$ to state-of-the-art algorithms. **Bold** values indicate a value below the significance level equal to 0.05, in favor of Alert$^+$. Underline values indicate p-values below original significance level, but not below the Holm's (Holm, 1979) corrected value, that depends on the number of tested hypothesis.

| $\alpha \rightarrow$
Alert$^+$ vs. $\downarrow$ | 0 | 0.1 | 0.2 | 0.3 | 0.4 | 0.5 | 0.6 | 0.7 | 0.8 | 0.9 | 1 |
|---|---|---|---|---|---|---|---|---|---|---|---|
| Economy | 1.0 | 0.759 | 0.034 | 0.083 | 0.024 | 0.049 | 0.016 | **<1e-3** | **<1e-3** | **<1e-3** | **0.002** |
| Calimera | 1.0 | 0.58 | 0.041 | 0.342 | 0.006 | 0.029 | 0.009 | **0.004** | **<1e-3** | **0.002** | **0.001** |
| Stopping Rule | 0.208 | 0.129 | 0.086 | 0.41 | **0.015** | 0.05 | **0.002** | **<1e-3** | **<1e-3** | **<1e-3** | **<1e-3** |
| Proba threshold | 1.0 | 0.223 | 0.078 | **0.037** | **0.002** | **0.002** | **<1e-3** | **<1e-3** | **<1e-3** | **<1e-3** | **<1e-3** |
| Earliest | **<1e-3** | 0.347 | 0.176 | 0.327 | **0.014** | **0.006** | **<1e-3** | **<1e-3** | **<1e-3** | **<1e-3** | **<1e-3** |

## B.4 Scatter plots: other competitors

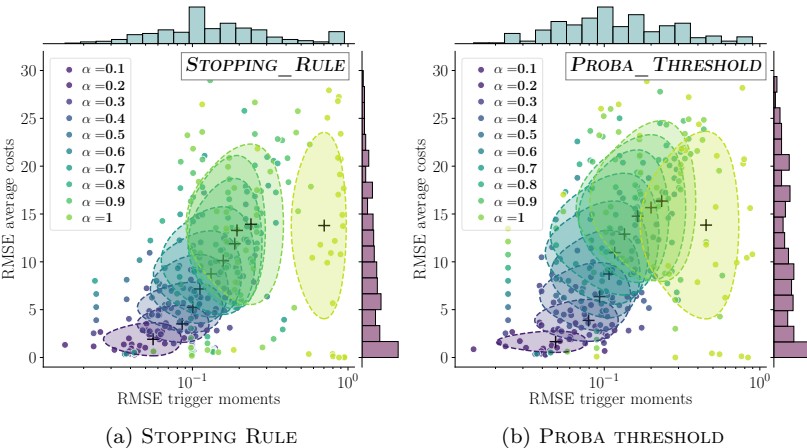

(a) Stopping Rule        (b) Proba threshold

Figure 8: The $x$-axis reports how far is the triggering time from the best a posteriori one: left is better. The $y$-axis reports the difference between the *AvgCost* incurred by the algorithm compared to the best a posteriori one, *AvgCost$^\star$* : lower is better. The black crosses report the average performance for each value of $\alpha$ (greater values correspond to higher relative importance of the delay cost). Ellipses display $2\times$ the standard deviation over both axis, computed for each $\alpha$ value.

## B.5 Linear/balanced cost setting

Alert is further compared with state-of-the-art methods under an alternative cost configuration characterized by a *linear* delay cost and a *balanced* misclassification cost, defined as $C_m(\hat{y} \mid y) = \mathbb{1}(\hat{y} \neq y)$ and $C_d(t) = \frac{t}{T}$. As illustrated in Figure 9, no significant performance differences are observed among the evaluated trigger models in this setting. These findings are broadly consistent with those reported by Renault et al. (2025), and highlight the importance of investigating alternative cost configurations to better discriminate between competing approaches.

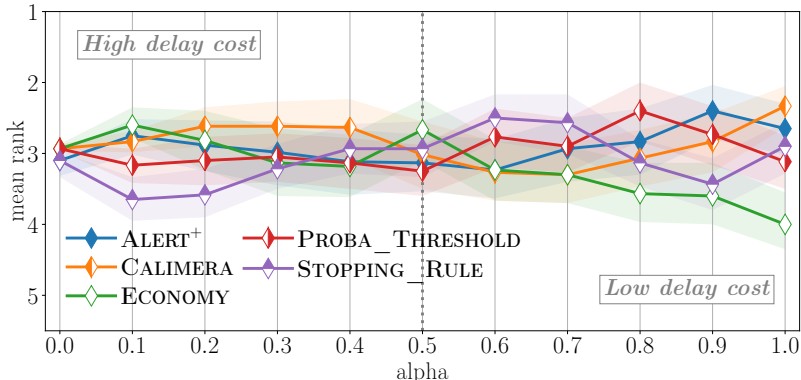

Figure 9: Evolution of the mean ranks, for every $\alpha$, based on the *AvgCost*. Linear delay cost and balanced, binary misclassification cost.

## C Runtime analysis

The runtime analysis presented in Table 4 shows that ALERT$^+$ requires substantially longer training times than competing methods. This additional computational cost arises from the regularization mechanisms introduced to stabilize the reinforcement learning process, which inevitably increase training overhead. However, once training is complete, ALERT$^+$ achieves inference speeds comparable to those of other approaches, ensuring that its deployment in real-world applications is not compromised.

Table 4: Computation times (in *sec.*) for $\alpha = 0.8$, five threads.

| Trigger | Median Fit | Median Inf. | Total |
|---|---|---|---|
| ALERT$^+$ | 123.057 | 0.011 | 9628.169 |
| CALIMERA | 5.459 | 0.026 | 177.848 |
| ECONOMY | 1.039 | 0.025 | 154.717 |
| PROBA_THRESHOLD | 0.052 | 0.024 | 5.305 |
| SR | 2.954 | 0.01 | 292.061 |

## D Sensitivity studies

### D.1 State space ablation

Each feature composing the ALERT$^+$ state space is systematically removed to assess its relative contribution. For $\alpha = 0.8$, Figure 10 indicates that removing the *time* information leads to the largest performance degradation, followed by the removal of the *maximum posterior*. The third most influential component is the *level of confidence*, which captures information about the distribution of maximum posterior probabilities over the training dataset. In contrast, the *margin* and the *predicted label* appear to contribute only marginally to performance in this setting.

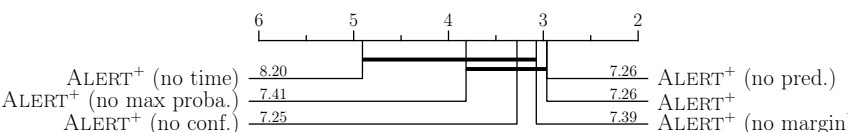

Figure 10: Wilcoxon signed-rank test labeled with mean *AvgCost*, $\alpha = 0.8$.

### D.2 Reward function

As outlined in Section 3.3, there exist multiple choices for the reward function. Since ECTS is a new problem for RL, it is important to assess whether some choices lead to significantly better results or if the reward function has little impact. One debate is whether it is better to postpone the delay cost signal to the end of an episode at the triggering time or if it should be distilled at each time step. Another consideration is which delay cost signal should be provided during learning. Specifically, should it incorporate information about the *a posteriori* best decision time $t^\star$?

To answer these questions, the following reward functions have been tested in our experiments using the same datasets as previously stated:

$$r(t) = \begin{cases} -\Delta C_d(t) & \text{if } a_t = \text{``wait''} \\ -C_m(\hat{y}_t|y) - \Delta C_d(t) & \text{if } a_t = \text{``trigger''} \end{cases} \tag{R1}$$

$$r(t) = \begin{cases} -\mathbb{1}(t > t^\star)\Delta C_d(t) & \text{if } a_t = \text{``wait''} \\ -|C_m(\hat{y}_t|y) - C_m(\hat{y}_{t^\star}|y)| & \text{if } a_t = \text{``trigger''} \\ \quad -\mathbb{1}(t > t^\star)\Delta C_d(t) \end{cases} \tag{R2}$$

$$r(t) = \begin{cases} 0 & \text{if } a_t = \text{``wait''} \\ -|t - t^\star| & \text{if } a_t = \text{``trigger''} \end{cases} \tag{R3}$$

For both reward functions R1 and R2, alternate versions in which all costs are given only at trigger time (i.e. end of episode), denoted with the "END" suffix, are tested. For all the experiments carried out, the ALERT$^+$ state space is kept constant and equal to: $\mathcal{S} = \{$*max posterior*, *margin*, *pred. class*, *level of confidence*, *time*$\}$.

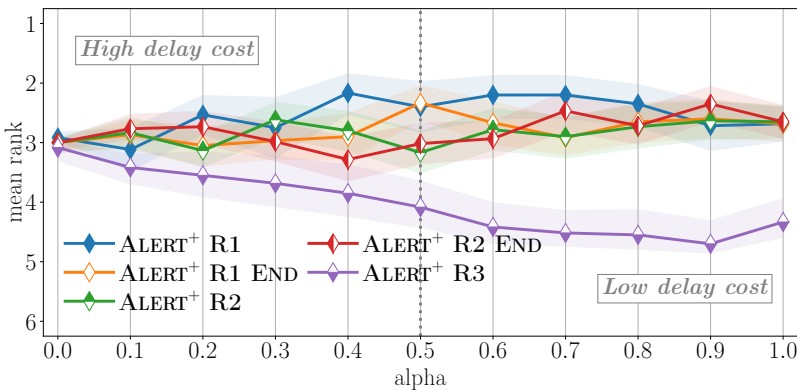

Figure 11: Robustness study on ALERT$^+$ reward's function.

Figure 11 synthesizes the results by comparing the ranking of five versions of ALERT$^+$ using the alternative reward functions. The first conclusion is that not providing the misclassification cost (ALERT$^+$ R3) leads to uniformly bad performance. The second one is that no other reward function can be singled out as the best one. In particular, no difference exists between providing the reward signal only at the end of each learning episode, or distilling the delay cost at each time step, even though ALERT$^+$ R1 seems a little bit better for intermediate values of $\alpha$. Similarly, information about $t^\star$ does not provide an advantage. This is consistent with the findings of figure 6a that shows that ALERT$^+$, the best performing algorithm, tends to move away from $t^\star$ to achieve better *AvgCost*s.

It thus appears that using the reward function R1 is a good choice.

### D.3 Discount factor $\gamma$

When varying the discount factor $\gamma$, ALERT$^+$ generally exhibits a tendency to postpone its decisions. This behavioral shift leads to degraded performance for low values of $\alpha$, corresponding to settings with high temporal costs. In contrast, when the temporal cost is less pronounced (top-right region of Figure 12), ALERT$^+$ remains close to the Pareto front even for low values of $\gamma$. Notably, for $\alpha = 1$, the configuration with $\gamma = 0.8$ outperforms the original ALERT$^+$, highlighting the potential regularization effect of $\gamma$ in regimes characterized by low temporal pressure.

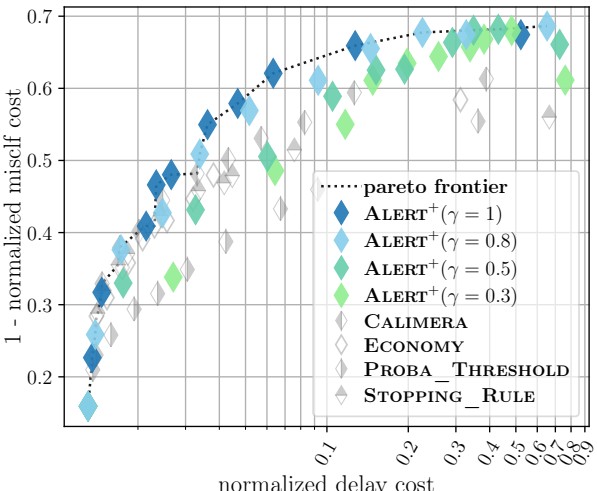

Figure 12: Pareto front, displaying for each $\alpha$, the normalized version of the *AvgCost*, decomposed over delay and one minus misclassification cost on $x$-axis and $y$-axis respectively. Best approaches are located on the top left corner. High $\alpha$ values are located on the right, low ones on the left. Due to the exponential shape of the delay cost, the $x$-axis is on log scale.

### D.4 DQN size

In this section, we evaluate several DQN architectures for ALERT$^+$. To limit computational overhead, experiments are conducted solely with $\alpha = 0.8$. As shown in Figure 13, increasing the number of layers and/or the hidden dimension does not yield significant performance improvements. Notably, the lowest *AvgCost* is achieved by the simplest architecture, consisting of a single hidden layer with 32 units, i.e. base choice for ALERT$^+$.

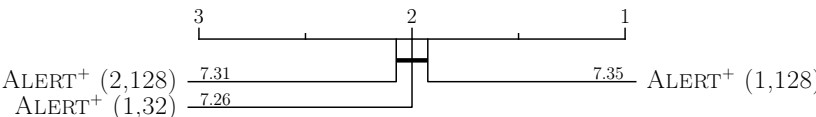

Figure 13: Wilcoxon signed-rank test labeled with mean *AvgCost*, $\alpha = 0.8$. DQN architecure is denoted as (num. of layers, hidden dim.).

### D.5 Sampling ratio

We further evaluate the robustness of the results with respect to the sampling frequency of the time series. In this experiment, finer-grained sampling resolutions, specifically every 1% and 2% of the time series, are

considered for $\alpha = 0.8$. As shown in Figure 14, ALERT$^+$ continues to significantly outperform state-of-the-art methods, and the overall ranking of the approaches remains globally consistent across different sampling resolutions. The only notable change is that PROBA THRESHOLD, although still ranked last, is no longer significantly different from the other state-of-the-art methods.

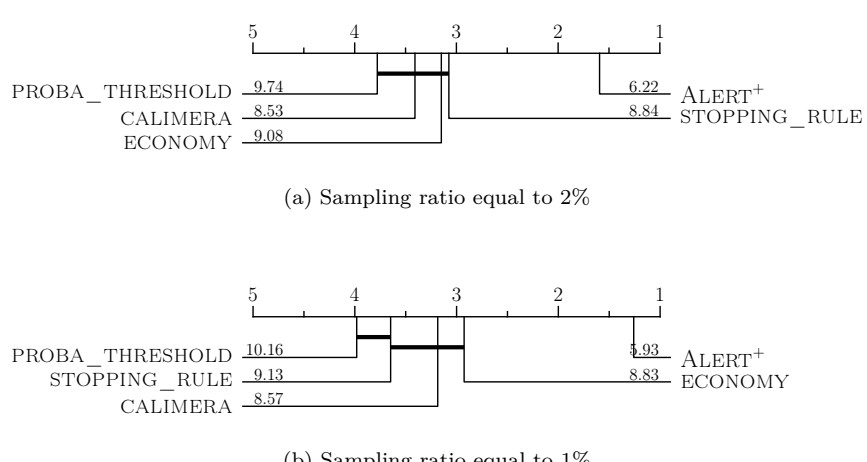

(a) Sampling ratio equal to 2%

(b) Sampling ratio equal to 1%

Figure 14: Wilcoxon signed-rank test labeled with mean *AvgCost*, $\alpha = 0.8$.

### D.6 Classifier

As an alternative to MiniROCKET for feature generation, Catch22 Lubba et al. (2019) and SAX Lin et al. (2007) are evaluated. Figure 15 shows that the overall ranking of the methods is only marginally affected by the choice of feature extractor, with ALERT$^+$ continuing to outperform state-of-the-art competitors.

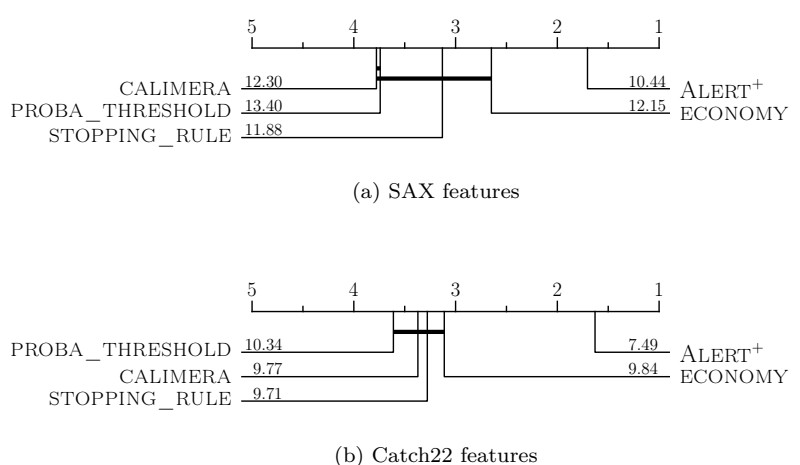

(a) SAX features

(b) Catch22 features

Figure 15: Wilcoxon signed-rank test labeled with mean *AvgCost*, $\alpha = 0.8$.

## E Qualitative results

### E.1 Visualizing trigger scores

In this section, we provide a visual interpretation of the strong performance achieved by ALERT$^+$. To this end, the trigger scores $\{score_t\}_{(1 \leq t \leq T)}$ produced by several trigger models are visualized alongside the true

and expected costs, the latter reflecting the confidence of the classifier. For each approach, the trigger score is normalized such that a decision is triggered whenever the score becomes strictly positive. In particular, the scores are computed as follows:

- ALERT$^+$: $score_t = Q(s_t, a_t = 1) - Q(s_t, a_t = 0)$, corresponding to the difference between the two $Q$-values.

- CALIMERA: $score_t = f_t(X_t)$, corresponding to the output of the regressor specialized for time step $t$.

- ECONOMY: $score_t = \min_{(t+1 \leq t' \leq T)} \hat{C}(t') - \hat{C}(t)$, defined as the difference between the minimum forecasted cost and the estimated cost incurred at the current time step.

- PROBA_THRESHOLD: $score_t = \max_{y \in \mathcal{Y}} p(y|\mathbf{x}_t) - \hat{\delta}$, representing the difference between the maximum posterior probability and the learned threshold.

- STOPPING_RULE: $score_t = \hat{\gamma}_1 p_1 + \hat{\gamma}_2 p_2 + \hat{\gamma}_3 \frac{t}{T}$, corresponding to the SR criterion as is.

Then, all of these scores are divided by the absolute maximum to get all of them in the $[-1, 1]$ range. For eight of the benchmark datasets, on which ALERT$^+$ outperforms its competitors, we plot the top three test instances exhibiting the largest difference in incurred cost between ALERT$^+$ and the average performance of competing methods. We set $\alpha = 0.8$, corresponding to the trade-off value for which ALERT$^+$ achieves its best performance. Figures 16–23 generally illustrate scenarios in which the cost-optimal decision points are narrow and therefore difficult to identify. In these cases, ALERT$^+$ demonstrates a stronger ability to capture complex patterns in classifier uncertainties and errors, which helps explain its superior performance relative to state-of-the-art approaches.

In Section E.2, the same examples are revisited using a higher sampling frequency, with a classifier trained at each 1% increment of the time series. This setting ensures that the original sampling rate of 5% is not overly coarse, which could otherwise lead to overly punctual cost optima, thus artificially favoring ALERT$^+$ against competing approaches. As shown in Figures 24–31, the superior capabilities of ALERT$^+$ are confirmed, and the initial analysis is further strengthened. Indeed, several examples exhibit substantially more complex patterns under this finer temporal discretization, yet ALERT$^+$ continues to handle these scenarios effectively. These examples are thus consistent with results displayed in Section D.5.

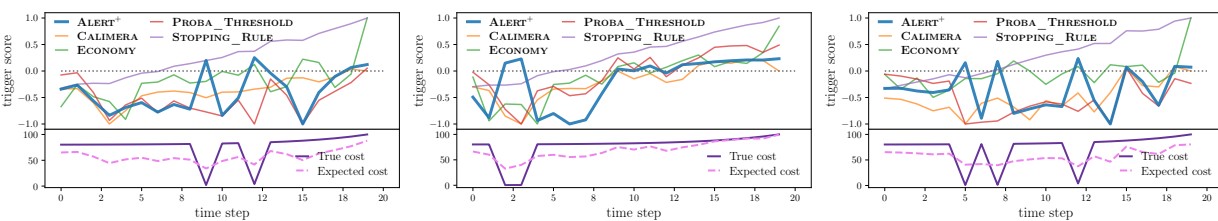

Figure 16: AluminiumConcentration

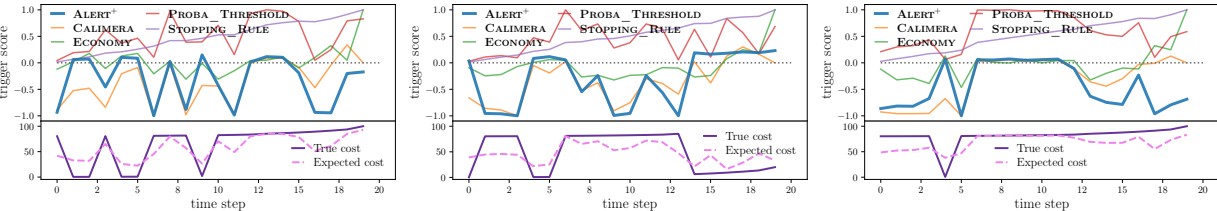

Figure 17: DhakaHourlyAirQuality

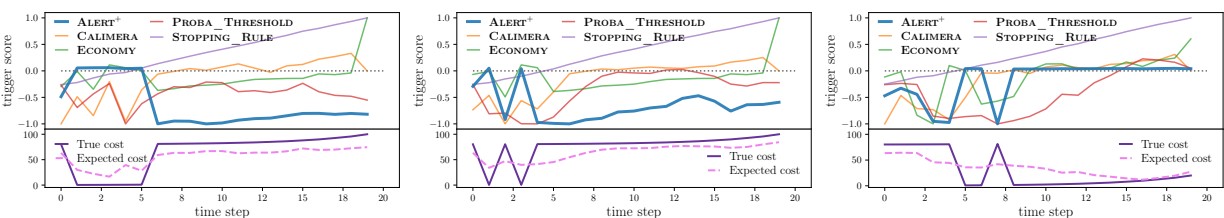

Figure 18: GunPointAgeSpan

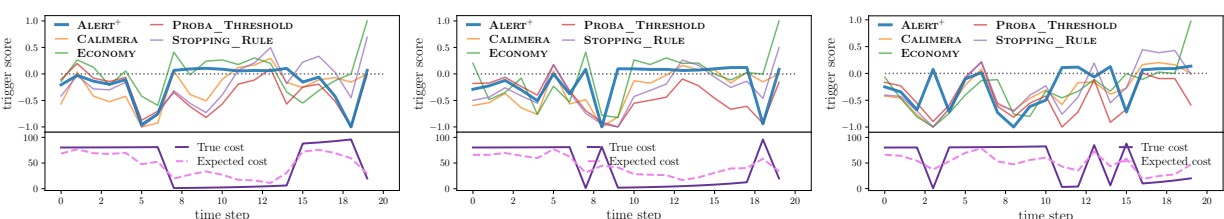

Figure 19: HouseholdPowerConsumption1

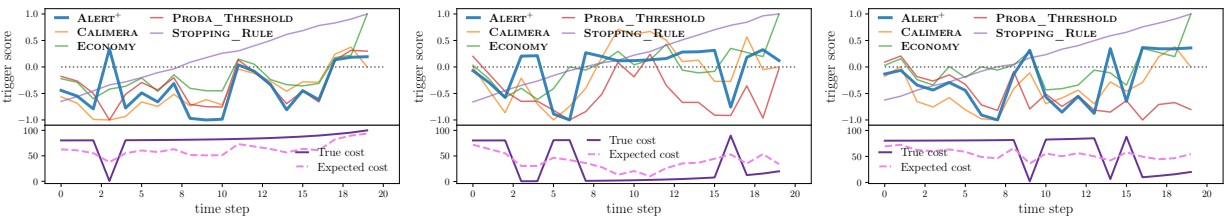

Figure 20: MadridPM10Quality

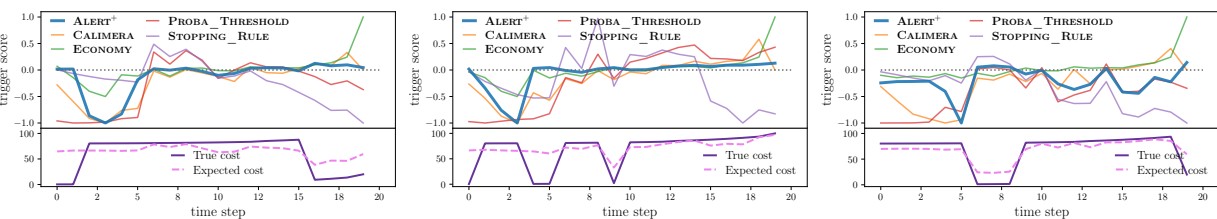

Figure 21: MelbournePedestrian

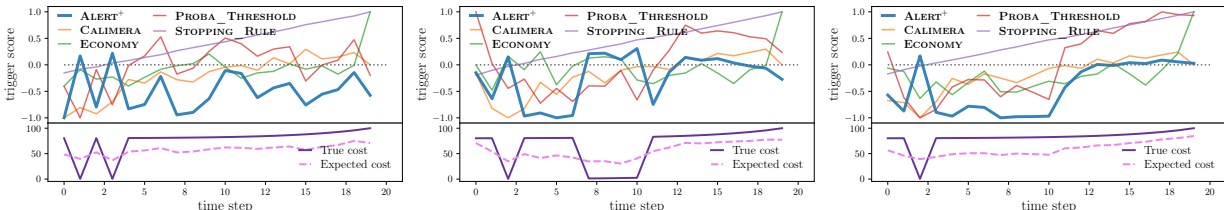

Figure 22: PrecipitationAndalusia

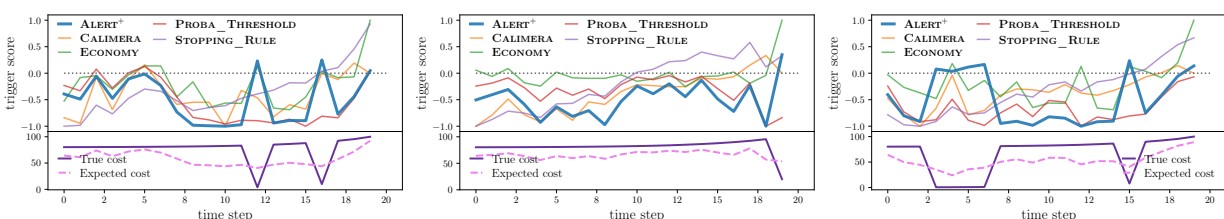

Figure 23: WindTurbinePower

## E.2 Robustness to sampling frequency

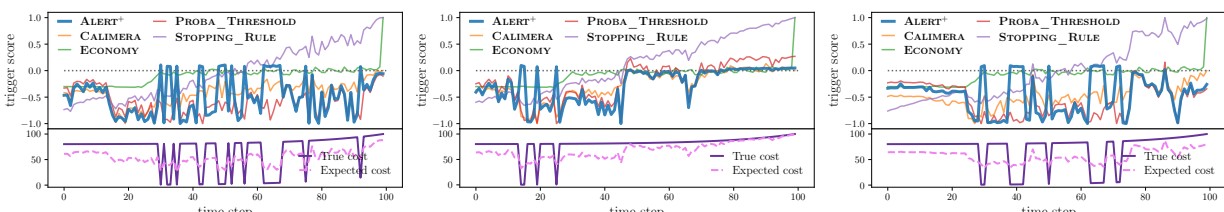

Figure 24: AluminiumConcentration

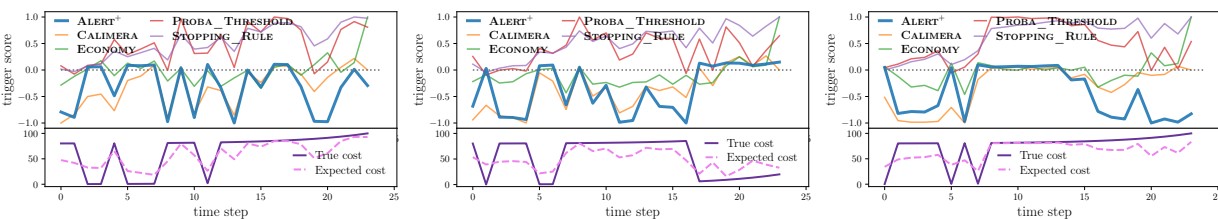

Figure 25: DhakaHourlyAirQuality

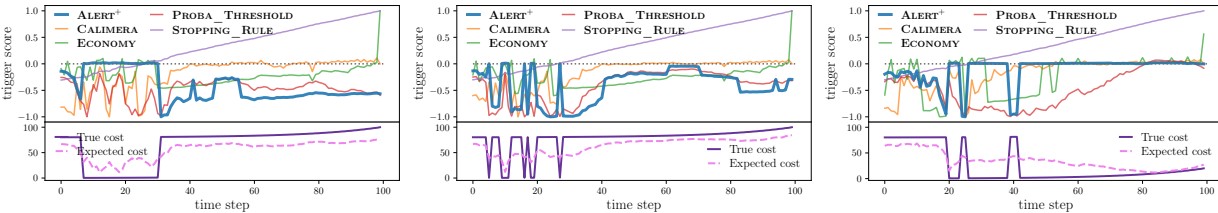

Figure 26: GunPointAgeSpan

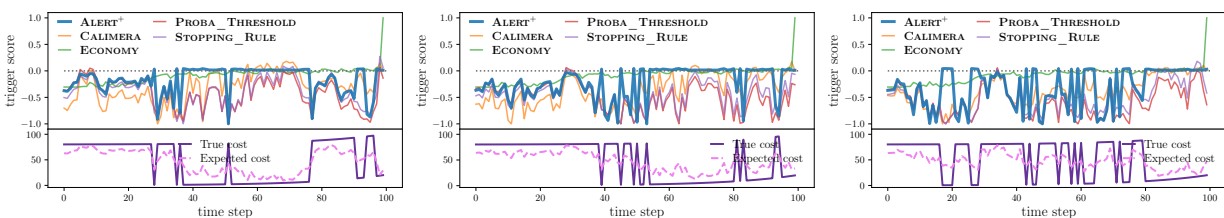

Figure 27: HouseholdPowerConsumption1

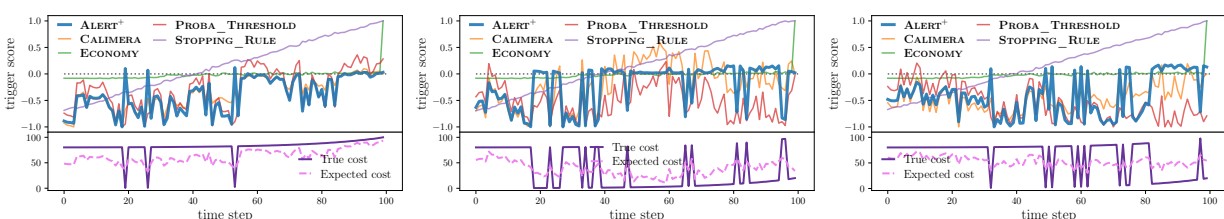

Figure 28: MadridPM10Quality

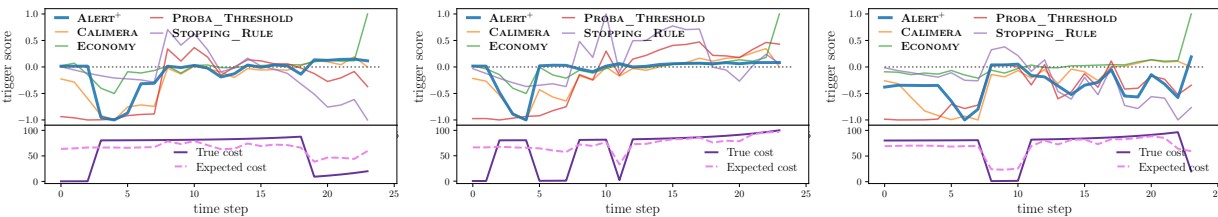

Figure 29: MelbournePedestrian

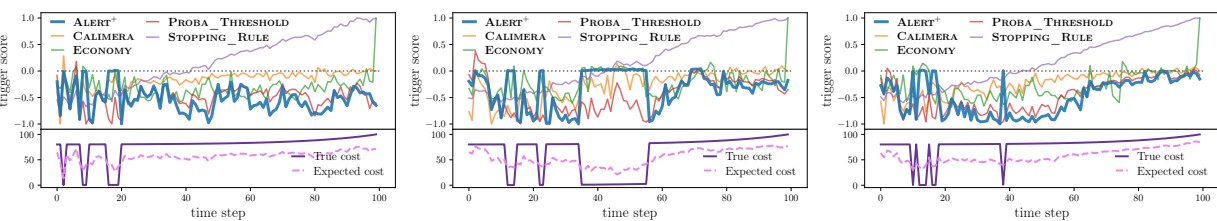

Figure 30: PrecipitationAndalusia

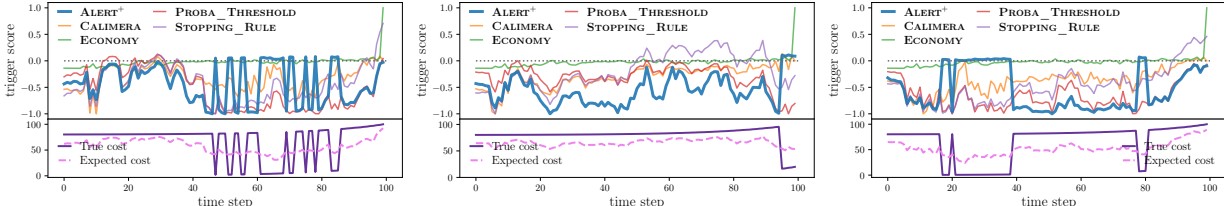

Figure 31: WindTurbinePower

