# OpenReview forum: "Alert: Learning Trigger Functions  for  Early Classification of Time Series using Deep-RL"
_TMLR — Under review for TMLR_

### Review · Reviewer_so54 · 2026-07-11

**Summary Of Contributions:**

This paper studies early classification of time series, where a model must decide not only what label to predict but also when to stop observing and make the prediction. The authors propose Alert, a classifier-agnostic RL-based trigger function trained with offline DDQN. The key idea is to keep the time-series classifier fixed and learn only the stopping policy from classifier outputs and timing/cost features. The strongest variant, Alert+, combines posterior probabilities, maximum confidence, margin, predicted class, confidence level, and time ratio as the trigger state.

The main strength is that the paper isolates the trigger-learning problem cleanly and compares RL triggers against several hand-designed separable ECTS methods under the same classifier setting. The experiments are fairly extensive, covering many datasets, cost trade-offs, state representations, reward variants, and classifier feature extractors. The main weakness is that the method is algorithmically simple, and much of the gain seems tied to a carefully chosen state representation and a particular asymmetric/exponential cost setting. The paper should be clearer about this scope.

**Audience:**

Yes

**Audience Explanation:**

Early time-series classification is a relevant topic for TMLR readers interested in time series, sequential decision-making, cost-sensitive prediction, and practical RL. The paper’s modular framing is useful: many real systems may already have a good classifier, and the harder deployment question is when to issue a prediction. The empirical finding that a learned trigger over compact classifier-derived features can outperform standard hand-designed triggers is likely to be useful to some readers.

**Claims And Evidence:**

Yes

**Claims Explanation:**

The central empirical claim, that Alert+ can outperform existing separable trigger functions under the paper’s benchmark setup, is reasonably supported. The authors compare against several relevant ECTS baselines, evaluate across many datasets and cost trade-offs, and provide ablations showing that state representation matters.

That said, the evidence supports a somewhat bounded claim. Alert+ is strongest under the paper’s main cost setting with imbalanced misclassification costs and exponential delay costs. The advantage appears weaker or not significant under more balanced/linear settings. Also, the results suggest that RL is not automatically better than handcrafted triggers given the same weak state space; the benefits emerge when the state representation is rich enough. The paper should state this more directly.

**Requested Changes:**

Critical for acceptance:

- Clarify the scope of the claims. The paper should avoid implying that RL triggers are generally superior to handcrafted triggers. The results seem to support a narrower claim: Alert+ works well when given a rich state representation and under the main imbalanced/exponential cost setting.

- Discuss the balanced/linear cost results more explicitly. If differences are not significant in these settings, this should be highlighted in the main text, not treated as a minor appendix observation.

- Add or expand analysis on calibration sensitivity. Alert+ relies heavily on posterior probabilities, confidence, and margins. A simple experiment with miscalibrated probabilities, temperature scaling, or calibration errors would help show whether the learned trigger is robust.

- Provide a more direct compute/training cost comparison. Alert+ appears more expensive to train than simple threshold or rule-based triggers. The paper should discuss when this cost is justified.

Would strengthen the work:

- Add a distribution-shift experiment, for example changed class priors or shifted time-series patterns at test time.

- Separate the contribution of state design from the contribution of DDQN more clearly. A strong non-RL classifier over the same Alert+ features would be a useful baseline.

- Include a few failure cases where Alert+ performs worse than simpler methods, since this would help readers understand when the method should not be used.

---

### Review · Reviewer_sADa · 2026-07-20

**Summary Of Contributions:**

The paper focuses on the Early Classification of Time Series (ECTS) using a separable, classifier-agnostic architecture. Its core contributions include:

- **Methodological Framework:** Introducing ALERT, an offline DDQN framework that operates entirely downstream of pre-trained, uncoupled classifiers. This represents a distinct departure from existing end-to-end or tightly coupled RL-based ECTS methods.
- **Empirical Analysis of Feature Space:** Providing a systematic exploration of alternative state representations across 30 public benchmarks to isolate what input configurations drive policy optimality.
- **The ALERT+ Model:** Proposing a robust variant utilizing a combined subset of state-of-the-art features that achieves superior mean rank across multiple cost weight regimes.

However, while the paper is well-organized and supported by strong reproducibility efforts, there are significant gaps between the authors' headline claims and the empirical evidence provided. See details in **Explain your answer above**.

**Audience:**

Yes

**Audience Explanation:**

ECTS is an active and growing subfield, as evidenced by dedicated benchmarks recently published in TMLR (Renault et al., 2025). Regardless of how the core empirical claims regarding RL superiority are ultimately scoped down, this submission offers several findings that are of clear, independent interest to the TMLR community:
- **A Modular, Classifier-Agnostic Testbed:** By explicitly decoupling the time-series classifier from the decision policy, the framework provides a clean experimental architecture that allows researchers to isolate whether performance gains stem from improved predictor representations or superior decision logic.
- **Actionable Insights on Feature Informativeness:** The systematic empirical analysis demonstrates that feature informativeness dominates triggering policy complexity, framing state-space selection as a primary, tunable hyperparameter—a valuable lesson for practitioners designing early decision systems.
- **Reproducibility and Utility:** The open-source repository aids reproducibility. Although the code requires further polish and structure before it can be fully utilized as a standard benchmark testbed, it holds strong potential for future research applications.

**Broader Impact Concerns:**

The methods developed in this paper facilitate decision-making under time constraints across critical domains, including medical diagnostics, industrial anomaly detection, and safety monitoring. By improving the efficiency and transparency of trigger policies, this work can assist practitioners in issuing timely, cost-sensitive alerts. However, deploying reinforcement learning-based triggering mechanisms in real-world environments entails non-trivial risks. If a policy triggers prematurely due to unmodeled noise or distributional shift, it may cause false alarms or critical misclassifications. The authors should include **Broader Impact Statement** that can address the above concerns.

**Claims And Evidence:**

No

**Claims Explanation:**

- **Disconnect Between Table 2 (Equal Information) and Narrative Claims:** The text states that \empth{"RL agents can match or improve upon strong handcrafted trigger functions"} and that \empth{"as the state space gets enriched, RL-based algorithms tend to outperform more significantly their handcrafted counterparts"}.  However, Table 2 (which tests p-values of AvgCost differences under matched feature sets) shows that: When restricted to small feature sets (e.g., ECONOMY vs. ALERT_Eco), the handcrafted method achieves statistically significantly lower AvgCost than its RL counterpart across $\alpha \in [0.4, 0.9]$ (bolded p-values $< 0.005$).  The equal-information RL variants do not achieve statistically significant wins over their handcrafted counterparts in Table 2 (italicized p-values). Hence, the reviewer conjecture that the performance gains demonstrated in Section 5.2 do not stem from RL superiorly executing decision rules on a given feature set; rather, they stem from ALERT+ operating on a much larger, combined feature set ($\mathcal{S} = \vert{}\mathcal{Y}\vert{}+4$). The authors should clarify this distinction so readers do not incorrectly conclude that RL inherently outperforms handcrafted rules when given the exact same restricted inputs.
- **Empirical Gap in the "Classifier-Agnostic" Claim:** The abstract and introduction emphasize that ALERT is a generic, classifier-agnostic framework capable of operating on top of arbitrary pre-trained classifiers. However, in the main experimental evaluation (Section 5), all experiments rely exclusively on MINIROCKET as the base classification backbone. While Appendix D.6 evaluates alternative feature extractors (SAX and Catch22), these represent feature representations rather than fundamentally different neural or ensemble model architectures. Different classifiers (e.g., Deep Recurrent/Transformer models vs. Gradient Boosted Trees) exhibit distinct calibration profiles, overconfidence tendencies, and uncertainty drift over time, which could affect the trigger agent's generalization.
- **Realistic Delay Modeling vs Early Signal Readiness:** The delay cost in Equation (8) is defined as a strictly monotonically increasing exponential function $C_d(t) = \exp\left(\frac{t}{T} \times \log 100\right)$. This formulation enforces a strong global assumption that earlier predictions are strictly cheaper and always preferred. In many real-world domains—such as clinical patient monitoring or industrial anomaly detection—early observations may contain mostly noise, providing negligible predictive signal until a critical physiological or operational transition occurs. Forcing an exponential delay penalty risks biasing the agent toward prematurely committing before sufficient signal has manifested. This concern is reinforced by Appendix B.5 (Figure 9), where the performance gap between ALERT+ and baselines mostly vanishes under a linear/balanced delay cost profile.

**Requested Changes:**

- Provide a supplementary experiment on a representative subset of datasets (e.g., 3–5 datasets) using a structurally distinct classifier backbone (such as a 1D-CNN, LSTM, or Transformer-based classifier) to demonstrate that the learned policy generalizes well across different probability calibration profiles.
- Discuss how the framework would handle non-monotonic delay functions or critical observation windows (e.g., where delay cost is flat/zero up to a threshold $t_{\text{target}}$  and spikes thereafter), where triggering "too early" is uninformative or undesirable.
- While the trigger policy itself is trained using DDQN, the inputs to the state space $\mathcal{S}$ remain largely handcrafted based on prior ECTS heuristics (e.g., margins, confidence bins, max posteriors). While this is acceptable for demonstrating that RL extracts better non-linear decision boundaries than traditional rule-based triggers under identical feature sets, the framework still relies heavily on manual state engineering rather than learning representations end-to-end. Furthermore, relying on static scalar summaries from probability distributions may limit the policy's ability to capture subtle temporal trends in classifier uncertainty across sequential time steps t. The reviewer suggest the authors to clarify in Section 6 (Conclusion/Discussion) that relying on pre-engineered scalar state features represents a conceptual boundary of the current implementation and discuss exploring dynamic representation learning and adopting constrained RL formulations as future work.
- Tone down assertions claiming that RL triggers "systematically outperform" or "improve upon" handcrafted rules when given identical inputs. Explicitly clarify in Section 5.1 that under matched feature spaces, RL-based triggers perform comparably to traditional handcrafted methods (and are sometimes significantly outperformed, as shown in Table 2 for ECONOMY).
- Clearly distinguish between the performance of equal-information RL variants (Table 2) and ALERT+. State transparently that ALERT+'s empirical gains in Section 5.2 stem primarily from feature enrichment ($\mathcal{S} = \vert{}\mathcal{Y}\vert{} + 4$) rather than a fundamental superiority of temporal-difference RL logic over rule-based decision mechanisms alone.

---

### Review · Reviewer_yrwV · 2026-07-21

**Summary Of Contributions:**

This paper introduces Alert, an offline Double DQN framework for learning trigger functions in separable and classifier-agnostic Early Classification of Time Series. Its main contribution is a controlled comparison in which RL-based and handcrafted triggers receive the same classifier-derived information. Experiments on 30 datasets systematically study the impact of state representation and motivate Alert+, which combines several standard ECTS features and achieves the best average rank across the tested cost trade-offs.

The main strengths are the fair comparison protocol, the systematic state-space analysis, and the extensive sensitivity studies. The main weaknesses are the limited algorithmic novelty of the RL method, some overly strong statistical claims, several unclear or inconsistent mathematical definitions, and the absence of a simple non-RL learning-based baseline using the same Alert+ state representation.

**Audience:**

Yes

**Audience Explanation:**

The paper should be relevant to researchers working on early decision-making, time-series classification, optimal stopping, cost-sensitive learning, and reinforcement learning. In particular, the controlled comparison between learned and handcrafted trigger functions is methodologically useful, while the state-space experiments provide practical insights into which classifier-derived signals are most useful for early triggering decisions.

Although the RL algorithm itself is standard, the paper offers a careful empirical investigation and a reusable framework that should be of interest to at least part of the TMLR audience.

**Broader Impact Concerns:**

The work does not raise major ethical concerns by itself.

**Claims And Evidence:**

Yes

**Claims Explanation:**

Overall, the main empirical claims are supported by extensive experiments, including comparisons on 30 datasets, statistical tests, state-space ablations, Pareto analysis, and sensitivity studies over rewards, discount factors, network sizes, sampling resolutions, and classifiers.

However, several claims should be qualified. Alert+ achieves the best average rank across all tested values of α, but statistically significant superiority after Holm correction is not observed against every competitor for every α. The strongest and most consistent significance is concentrated in the higher-α regime. In addition, the current experiments do not fully determine whether the gains come specifically from reinforcement learning, rather than from the richer Alert+ state representation and nonlinear function approximation.

Thus, the main empirical conclusions are convincing, but some statements about universal superiority, Pareto dominance, and the specific benefit of RL should be moderated or supported by additional evidence.

**Requested Changes:**

- The main contribution should be positioned as the separable, classifier-agnostic comparison framework and the systematic state-space study, rather than as a major RL algorithmic contribution.
- The paper should distinguish between best average rank and statistically significant superiority. Claims that Alert+ significantly outperforms all competitors across all α should be revised. Table 2 should also use a multiple-testing correction consistent with Table 3, or the difference should be justified. The NaN result for Economy at α=0 should be explained.
- Equation (2) appears to have an incorrect sign, and Equation (10) has inconsistent summation bounds. The paper should also explain how α is incorporated during training, reward computation, validation, and model selection, and define the normalization used in Figures 2 and 5.
- The authors should clarify what happens if the policy continues to wait until T, how terminal transitions are represented, why γ=1 is used by default, and whether the chosen state representation is assumed to be strictly or approximately Markov.
- A supervised or imitation-learning trigger using exactly the same Alert+ state representation would help determine whether the gain comes from RL itself or from richer input features and nonlinear function approximation.